# Super-Resolution Images Methodology Applied to UAV Datasets to Road Pavement Monitoring

**Laura Inzerillo** , **Francesco Acuto \*** , **Gaetano Di Mino** and **Mohammed Zeeshan Uddin**

DIING—Department of Engineering, University of Palermo, Viale Delle Scienze Ed. 8, 90128 Palermo, Italy; laura.inzerillo@unipa.it (L.I.); gaetano.dimino@unipa.it (G.D.M.); mohammedzeeshan.uddin@unipa.it (M.Z.U.)
\* Correspondence: francesco.acuto@unipa.it

**Abstract:** The increasingly widespread use of smartphones as real cameras on drones has allowed an ever-greater development of several algorithms to improve the image's refinement. Although the latest generations of drone cameras let the user achieve high resolution images, the large number of pixels to be processed and the acquisitions from multiple lengths for stereo-view often fail to guarantee satisfactory results. In particular, high flight altitudes strongly impact the accuracy, and result in images which are undefined or blurry. This is not acceptable in the field of road pavement monitoring. In that case, the conventional algorithms used for the image resolution conversion, such as the bilinear interpolation algorithm, do not allow high frequency information to be retrieved from an undefined capture. This aspect is felt more strongly when using the recorded images to build a 3D scenario, since its geometric accuracy is greater when the resolution of the photos is higher. Super-Resolution algorithms (SRa) are utilized when registering multiple low-resolution images to interpolate sub-pixel information The aim of this work is to assess, at high flight altitudes, the geometric precision of a 3D model by using the the Morpho Super-Resolution™ algorithm for a road pavement distress monitoring case study.

**Keywords:** super-resolution of images; UAV survey; photogrammetry; algorithms; road pavement monitoring

## 1. Introduction

The use of the drone for monitoring the road pavement is becoming increasingly widespread to have continuous feedback on the health of the infrastructure and to be able to monitor the deviations that occur from one acquisition to another. However, when for reasons of physical constraints, one is forced to perform flights at high altitudes, the metric error of the final model is greater than 10 cm, which is unacceptable for monitoring purposes. The super-resolution imaging (SR) allows the shortcomings of the capture acquisition systems to obtain a higher-resolution image based on images which come from the same scene. It is an inverse problem, still not determined, for which it is necessary to provide a large amount of deep preliminary data in order to narrow the field of possible solutions. The super-resolution (SR) of the images, that allows to carry out a high-resolution image from a single low-resolution image, presents many solutions that can be associated to any given low-resolution pixel. It is an inverse problem that, nowadays, is not determined since it presents several solutions. To overcome this problem, it is necessary to provide deep preliminary information in order to constrain the space of the solution.

SR image is a typical problem in the computer vision field [1–11]. Nevertheless, light or geometric conditions i.e., long distance from the object during photogrammetric detections lead to problems in the construction of the 3D meshed model and therefore involve to unacceptable RMSE final value [12]. The creation of a 3D model by using historical photos of a no longer existing building constitutes another typical sample where the images are at a low resolution. But particular interest regards the UAV acquisition for the monitoring of

the state of road pavements condition [13]. The goal is to assess the 3D model's accuracy from a photogrammetric survey with super resolution images, and more specifically the validation of RMSE values close to a ground truth 3D in which any corrections have been provided [14] (Figure 1).

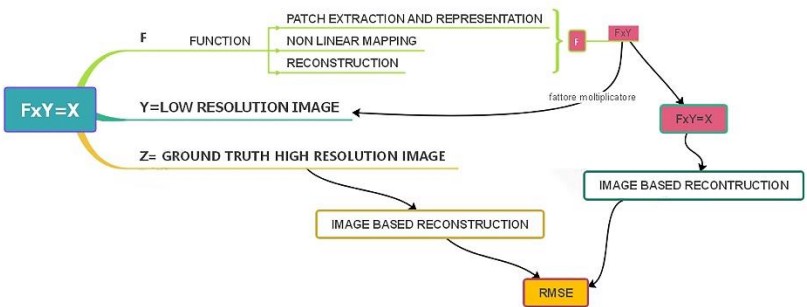

**Figure 1.** Proposed methodology overview.

For this purpose, a road pavement surface was chosen as sample test. In the case of flat surfaces, as in the case study, the transformation between image and object coordinates is obtained by planar homography [15], which is described by a $3 \times 3$ non-singular matrix **H**:

$$x = \begin{bmatrix} x \\ y \\ z \end{bmatrix} = \begin{bmatrix} h_1 & h_2 & h_3 \\ h_4 & h_5 & h_6 \\ h_7 & h_8 & h_9 \end{bmatrix} \begin{bmatrix} X \\ Y \\ Z \end{bmatrix} = \mathbf{HX} \tag{1}$$

where vectors $x = [x\, y_1]^T$ and $X = [X\, Y_1]^T$ express image and object points in homogenous coordinates, respectively. The GRS, in that case, has two degrees-of-freedom (DoF) only, given that the object is planar. Equation (1) shows two essential aspects: just one image allows the reconstruction of an object that is planar to be carried out, and since **H** is non-singular, the inverse transformation is always processed. On the other hand, object coordinates (X) are not always available. In this case, homography can be estimated with the Barazzetti proposal [16].

A known distance measured in the field allows the 3D model to be scaled [14], and the information to reconstruct the 3D coordinates is sufficient so that there is no need to acquire metric data such as object point or known ratios of distances and angles [17,18].

## 2. Related Works

The SRa problem has been mainly investigated in the Computer Vision field. The SRa have been verified and evaluated by Yang et al. in 2014 [19]. Among them, the example-based methods [20] achieved the state-of-the-art performance. Other studies proposed mapping functions such as kernel regression [21], simple 3 function [22], random forest [23] and anchored neighbourhood regression to further improve the mapping accuracy and speed. The sparse coding-based method and its several improvements [8,12,24] are among the state-of-the-art SR methods nowadays. In the aforementioned methods, the patches are the focus of the optimization; the patch extraction and aggregation steps are considered as pre/post-processing and handled separately.

The learning-based methods consist of algorithms which can be grouped into external, internal, and convolutional neural networks as main categories, depending on the source of the training dataset [25]. The learning algorithms, on which the external image super-resolution method is based, provide the relationship between low and high-resolution image patches. The aforementioned algorithms include several methods of which the convolutional neural network algorithm appears to be the most efficient [26]. This approach is usually performed on images with lots of patterns and textures, but it doesn't work well on the image structures outside the input image and it fails to generate a correct prediction on images of other classes [25,26].

About the deep learning techniques based on neural networks algorithms for image super-resolution, little research is currently available. Nevertheless, due to their reliability, the multi-layer perception (MLP) algorithms are used for blurring and for natural image denoising [27,28].

In a recent study, a super-resolution method has been proposed by Ahmadian et al., in which a second-order image gradient allows the edges and details of high-resolution and low-resolution images to be obtained [25]. A competitive learning is employed unlike other neural networks that use the backpropagation method for the weight vectors updating [25]. In particular the researchers used old and modern image datasets to train the single image super-resolution algorithm according to the self-organizing neural network, the "k-nearest" neighbour algorithm and Laplace gradient operator, comparing the performance with other previous works and showing that despite the potential of this approach, the processing speed of the algorithm is, however, slower than other traditional methods [25,29,30].

Concerning the aim of this paper, super-resolution reconstruction (SRR) technology, especially the convolutional neural network (SRCNN), has been shown to be appropriate for the detection of structural cracks based on images acquired using UAVs [31]. The performance of the computer vision crack detection models strongly depends on the quality of the images collected [32]. Even if UAVs are common and efficient equipment to collect images of concrete structures or road pavement surface, the vibrations and the distances to the target surveying area may generate problems which lead to the loss of image information and make it difficult to detect the cracks [31–35].

SRR algorithms can overcome the problems of motion blur and insufficient resolution of the images, improving the accuracy needed to detect surface distresses as cracks. Several methods based on deep learning are available to improve the SRR algorithms' performance, but few studies have employed these kinds of techniques focusing on the crack detection as a research objective. A comparison between the SR images reconstructed by the SrcNet® model with low-resolution images was performed by Bae et al., in which they showed that while SR images improve the recall of detection, at the same time a decrease in detection accuracy has been observed [36]. Other authors found that the crack segmentation accuracy improved with SRR for Low Resolution crack images, but the effect of the SR reconstruction on the quantification of crack features was not explored [37–39].

It is clear that the influence of several SRR networks on crack or surface distress reconstruction has not yet been fully investigated.

## 3. Methodology

### 3.1. Image Processing with Super-Revolution

Single image super-resolution (SISR) provides the opportunity to reconstruct a high-resolution image ISR from a single low-resolution image ILR. The relationship between ILR and the original high-resolution image IHR is variable relating to the different situations [40]. Several studies assume that ILR is a bicubic down -sampled version of IHR, but other factors of degradation, such as blur, decimation, or noise can also be considered for practical applications [41]. In order to achieve an ISR, the spatial resolution of images needs to be increased or simply the number of pixel rows/columns or both in the image. Several methods have been involved: the interpolation-based methods—*Image interpolation* (image scaling), that refers to *resizing digital images* and is widely used by image-related applications. Regarding traditional methods, they include: *Nearest-neighbour Interpolation*. The nearest-neighbour interpolation which is an algorithm able to select the value of the nearest pixel for each position to be interpolated regardless of any other pixels; *Bilinear Interpolation*—the bilinear interpolation (BLI) that provides better performance than nearest-neighbour interpolation while keeping a relatively fast speed and finally the *Bicubic Interpolation*—similarly, the bicubic interpolation (BCI) that performs cubic interpolation on each of the two axes compared to BLI, the BCI takes $4 \times 4$ pixels into account, and results in smoother results with fewer artifacts but much lower speed.

Starting from the high-resolution image, the low-resolution image is modelled using the expression shown below Equation (2) where X is the high-resolution image, Y is the low-resolution image, F is the degradation function and σ the noise.

$$X = F(Y; \sigma) \tag{2}$$

The degradation parameter σ is unknown; only the high-resolution image and the corresponding low-resolution image are provided. In order to find the inverse function of degradation, the neural network implementation could be involved, just using the SR and LR image data [42].

Learning the end-to-end mapping function F requires the estimation of network parameters $\Theta$ = {W1, W2, W3, B1, B2, B3}. That estimation allows the minimizing of the MSE between the reconstructed images F (Y; $\Theta$) and the high-resolution images X, used at the start. The algorithm developed Equation (3) includes Mean Squared Error as the loss function, starting from a series of high-resolution images {Xi} and the corresponding low-resolution ones:

$$L(\Theta) = 1 \cdot n \cdot X_{ni} = 1 \cdot ||F(Y_i; \Theta) - X_i||^2 \tag{3}$$

where n is the number of training samples. The strategy to consider MSE as the loss function offers a high PSNR that is a widely used metric for quantitatively evaluating image restoration quality, and it is related to the perceptual quality in partial quantity. Nevertheless, Chao Dong et al. [26] demonstrated that the application of Equation (4), where the weight matrices are calculated, proved to be performant; the loss is minimized using the stochastic gradient descent with the standard backpropagation.

$$\Delta_{i+1} = 0.9 \cdot \Delta_i - \eta \cdot \frac{\partial L}{\partial W_i^l}, \; W_i^l = W_i^l + \Delta_{i+1} \tag{4}$$

where $l \in \{1, 2, 3\}$ and i are the indices of layers and iterations, η is the learning rate, and $\frac{\partial L}{\partial W_i^l}$ is the derivative. The filter weights of each layer are initialized by drawing randomly from a Gaussian distribution with zero mean and standard deviation 0.001 (and 0 for biases). The learning rate is $10^{-4}$ for the first two layers, and $10^{-5}$ for the last layer. They empirically found that a smaller learning rate in the last layer is important for the convolutional neural network to converge [26].

The quality of the images, and the corresponding 3D model, significantly improves with the use of interpolation provided by the end-to-end mapping function.

*3.2. Photogrammetric Technique for Pavement Distress Detection Using Images Obtained by Drone*

Several studies have been conducted using photogrammetric techniques aimed at the pavement distress analysis. Nevertheless, the more considered aspect was the detection of cars or of vegetation. About the geometric accuracy aimed at the pavement distress detection there are no articles that research on the improvement of the source image quality. The experimentation that was conducted in this paper shows that, once the UAV photogrammetric survey has been made, it is possible to upgrade the resolution of the image source to achieve a more detailed 3D model.

A DJI MAVIC 2 Pro drone with the features shown in Table 1 was used (Figure 2).

To achieve good survey results, it is necessary to plan the flight path. Usually, when using a camera mounted on a drone, it is better to follow a serpentine path rather than a straight-line path. Nevertheless, it depends on the features of the road and on its edge. In our case, the flight was considered in a circle "hyperlapse" due the width of the road and the presence of light poles in the centre of the road, being careful to guarantee the photogrammetry overlap for the recognition of homologous points.

**Table 1.** UAV and camera settings for the surveys.

| Device | DJI Mavic 2 Pro | Camera [1] |
| --- | --- | --- |
| Camera resolution (megapixel) | 20 | 20.9 |
| Image size (pixel) | 5568 × 3648 | 5568 × 3712 |
| Sensor size (mm) | 13.2 × 8.8 | 23.5 × 17.5 |
| Focal length (35 mm eq.) | 28 | 24 |
| ISO | 200 | 100 |
| Shutter speed | 1/60 to 1/125 | 1/250 |
| Aperture | f/5.6 | f/8 |

[1] A Nikon Zfc mirrorless camera was used for the ground truth survey.

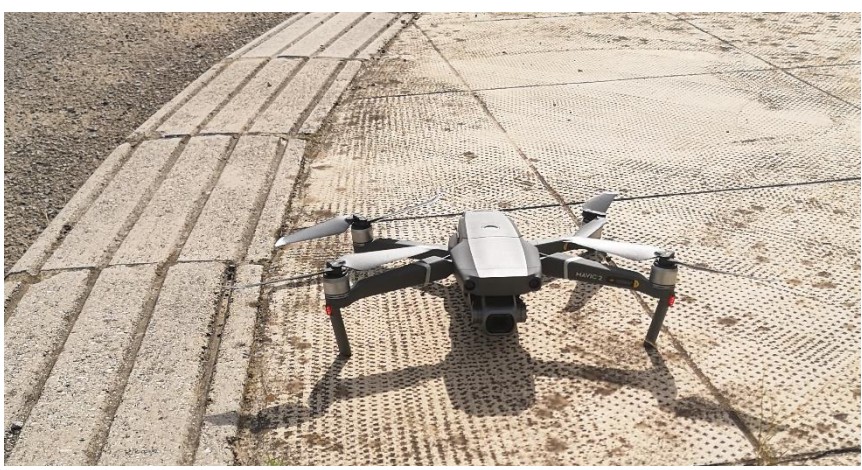

**Figure 2.** DJI MAVIC 2 Pro drone used for the surveys.

The most critical parameter to be considered is the ground sampling distance (GSD). The GSD is considered as a representation of the smallest details that can be accurately observed on an image; the smaller the value of the GSD, the greater the measurable details [17]. Models are interpreted from this parameter, since it has been demonstrated that the smallest visible details are two to three times the value of the GSD [43]. Based on manuals, generally, the smallest cracks and common distress are not smaller than 10 mm and with resolutions of 3 mm these distresses can be accurately identified. Therefore, appropriately using a 3 mm resolution for pavement distresses, the GSD should be no greater than 1 mm.

The GSD is given by the Equation (5) below, where D = object distance, $f$ = focal length, and pxsize = pixel size:

$$\text{GSD} = \frac{\text{D·pxsize}}{\text{f}} \tag{5}$$

The field of view (FOV) was 46.7°, calculated using Equation (6):

$$\text{FOV} = 2 \cdot \tan^{-1}\left(\frac{\text{d}}{2 \cdot \text{f}}\right) \tag{6}$$

where d is the diagonal length of the sensor and f is the focus length. Camera calibration was performed as part of the Structure from Motion (SfM) process, which calculated the initial and optimized values of the interior orientation parameters. The terrestrial images have been taken considering an altitude of 30 m.

## 4. Investigation

### 4.1. Road Pavement Monitoring

Road pavement distress detection and analysis is a crucial aspect for transportation authorities to optimize maintenance strategies. The distress evaluation represents one of the most important steps of the well-known pavement management system (PMS) analysis

method, which usually requires reliable measurements of the geometric characteristics of the damages [44–46]. In particular, the classification of distresses based on their severity classification needs very accurate measurements that the conventional surveying devices guarantee even if at high cost in terms of technologies and techniques [35].

In recent years, image-based technologies for automated distress detection have represented an assessed alternative option to the conventional ones [47–50]. High resolution imagery is central to efficiently detect and measure the road surface, even if the standard aerial imagery implies limitations in the survey handling and high costs [51]. For this reason, UAVs are increasingly employed to achieve high flexibility, lower costs, and quickness in the large-scale surveying field. UAVs allow images to be recorded with centimeter spatial resolution, providing a sufficient detail for detection and extraction of some pavement condition features, after being processed into the 3D reconstruction [51].

In several studies UAV image datasets have been used to reconstruct the road pavement surface, to observe its conditions, and, more specifically, to measure, in an accurate way, the deformations in distresses such as potholes and rutting [13,35,51–55].

In order to identify the severity of certain road surface distress, a cheaper methodology to process datasets from drone acquisitions is the stereovision approach, in which photogrammetry and structure-from-motion (SfM) are included [56]. The 3D model output allows to scan the desired metric information of the surveyed distresses efficiently. SfM can be used for pavement distresses such as rutting, block cracking, transverse cracking, potholes, and it enables the surveyor to match the requirements provided from the pavement distress manuals (Figure 3) [46]. This last aspect has prompted the present research work to consider the improvement in UAV image detection as the main goal to achieve the accuracy required from the international distress manuals (Table 2).

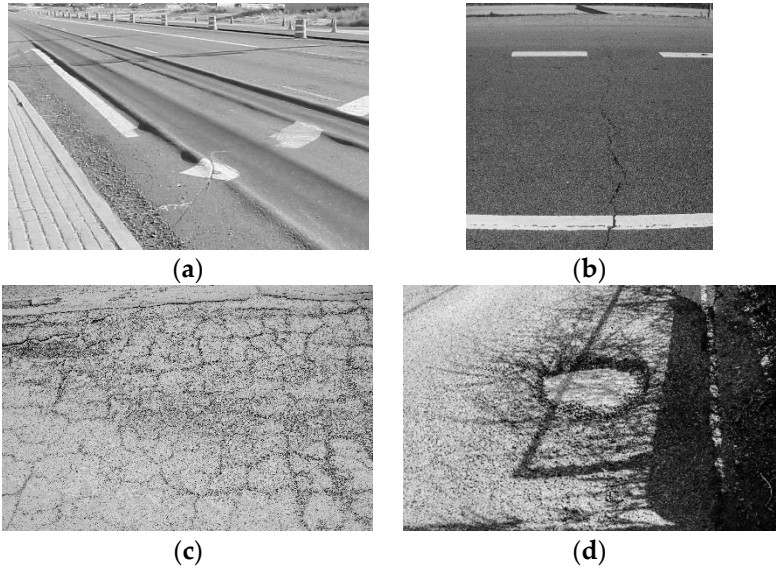

**Figure 3.** Road pavement distress examples: (**a**) rutting; (**b**) transverse cracking; (**c**) block cracking; (**d**) potholes.

**Table 2.** Road pavement distress manuals indications related to 4 considered distresses.

| Distress | Indicator | Severity Levels [1,2] |
|---|---|---|
| Block Cracking | Crack Width (mm) | 3–19 mm |
| Transverse Cracking | Crack Width (mm) | 3–19 mm |
| Potholes | Depth (mm) | 25–50 mm |
| Rutting | Depth (mm) | >12 mm |

[1] In the distress identifications manuals [46], for cracking phenomena, a low level is defined for width values less than 6 mm, a medium level for width values from 6 to 19 mm and a high severity level for width values greater than 19 mm. [2] Concerning potholes and rutting distresses, there is no distinction in severity levels.

At a certain flight altitude the spatial resolutions of the UAV images represent a limitation to the detection of distresses as individual cracks, given that most of their width is less than 0.01 m. Rather, considering a UAV device equipped with a CMOS sensor resolution 12 mpx which flies from 5 mt to 10 mt of altitude, it could be possible to generate a spatial resolution up to the millimeter order of accuracy [35].

To deeply investigate the accuracy of UAV images for road pavement monitoring, and the desirable improvement of them to generate precise 3D models, a case study in Palermo, Italy was performed implementing the previously mentioned Super-resolution approach.

### 4.2. Case Study

Concerning the present study, the developed algorithm has been used to demonstrate if the relative quality of the reconstructed 3D model of a road surface is sensitive to improvements in the images' quality and to quantify that improvement. A parking area within the University of Palermo, Italy was chosen as a test road to avoid the restrictions imposed on the drone flights (Figure 4) [57]. Two data sets were collected: a low-resolution image set and the other with high-resolution image, respectively. The low-resolution datasets have then been modified using the morpho super-resolution algorithm and compared with a ground truth source.

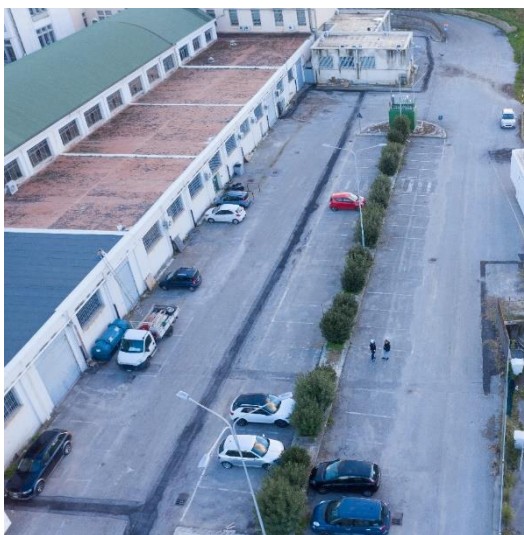

**Figure 4.** Low Image resolution of case study from drone camera.

The morpho super-resolution™ algorithm is supported by a self-titled software in which the following functions are available:

- *Video stabilization Movie Solid*, a technique named "electronic image stabilization" that cancels out camera shake electronically by cropping an area of an image. Another technique called optical image stabilization is provided to mechanically move the lens to compensate for the shaking of the camera [58]. The huge advantage of electronic image stabilization over optical stabilization is the absence of special hardware requirements, so that is sufficient to work using inexpensive products. The disadvantage is the shrinkage of the effective angle of view due to the image always being cropped.
- *Image Stabilization PhotoSolid*, that provides sharp images without camera shake or noise [59].Those who have a single-lens reflex camera may know well that camera shake and noise are the counterparts related with image degradation. Cameras, not limited to those of mobile phones, are devices that measure the amount of incident light. In other words, the more light enters a camera, the brighter the image is (and vice versa). When taking photos in a dark scene such as at night, noise is more prevalent than incoming light, which results in noisy images.

- *Image Enhancement by AI Based Segmentation and Pixel Filtering Morpho Semantic Filtering*™, which is an image enhancement software that implements AI based segmentation and pixel filtering [60].
- *Fast AI Inference Engine "SoftNeuro®"*, that operates in multiple environments, utilizing learning results that have been obtained through a variety of deep learning frameworks. It's user-friendly and it doesn't require any Deep Learning knowledge. SoftNeuro can also import from various frameworks and run fast on several and various architectures. It is both flexible and fast due to the separation of the layer and its execution pattern, which is a concept of routine.

Morpho super-resolution™ software provides a valid support to achieve the SRa starting from a LR and this is one of the most important features in image based modelling [61]. Once all of the images are converted from low-resolution to super-resolution, the image-based reconstruction can be deployed (Figure 5).

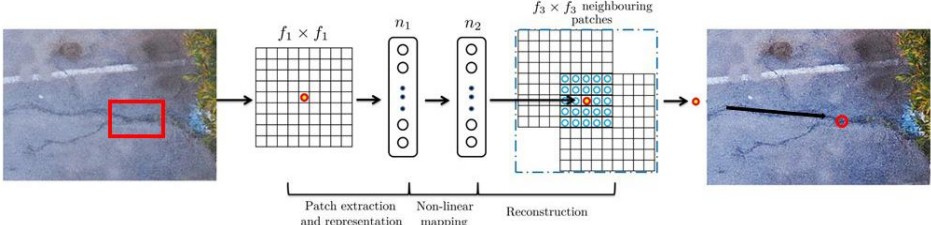

**Figure 5.** In the picture, there are two different steps in the processing to carry out the super resolution image from a low one: $f_1 \times f_1$ and $f_3 \times f_3$. Starting from a low -resolution image (Y), the first functional layer of the process extracts a set of feature maps. The last layer combines the predictions within a spatial neighbourhood to produce the final high-resolution image F(Y) called super-resolution Image. Between the two different layers, there is the non- linear mapping. This method is the sparse-coding-based one in the view of a convolutional neural network.

As previously mentioned, the three datasets to process are the low-resolution (drone output), super-resolution (drone outputs processed) and ground truth, respectively.

*4.3. SfM Reconstructions*

Within the length of the road, a limited area was chosen on which the distresses acquisition and processing was focused. After the image processing to convert low-resolution pictures into Super-resolution ones, the output dataset was used to build the points dense cloud of the road pavement surface by means of the Agisoft Metashape Pro software [62].

The same processing was implemented on all of the datasets [63] and, consequently, the RMS between the dense cloud of both dense clouds generated from dataset was assessed. The reliability of the ground truth processing was also verified [64]. Actually, the mentioned approach is widely used in computer vision for image detection but it is not common for SfM editing [65,66].

Secondly, to have a further validation of the methodology, the dense cloud built from low resolution images dataset, was elaborated [67,68]. In addition, a comparison between that dense cloud with the one obtained from the high-resolution images was carried out. In this way, it was possible compare RMS results from the low resolution image origin dense cloud, from the high resolution image source, and the super-resolution image dense cloud, obtained from the processing, and the high resolution source one [69].

In the following figures (Figures 6–9), the difference between the SfM reconstruction obtained from the different data set, the low and super resolution ones, are shown below.

CloudCompare v2 opensource software was used to investigate the quality of the 3D information [70]. The dense clouds, before their implementation in Cloud Compare platform, were scaled in Agisoft Metashape during the reconstruction step [71–73].

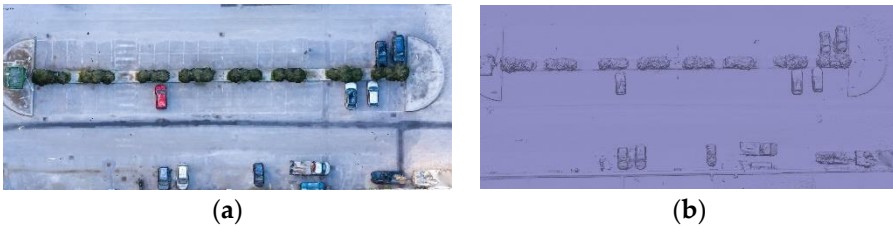

**Figure 6.** SfM reconstruction using low-resolution image dataset: (**a**) model shaded view; (**b**) solid mesh view.

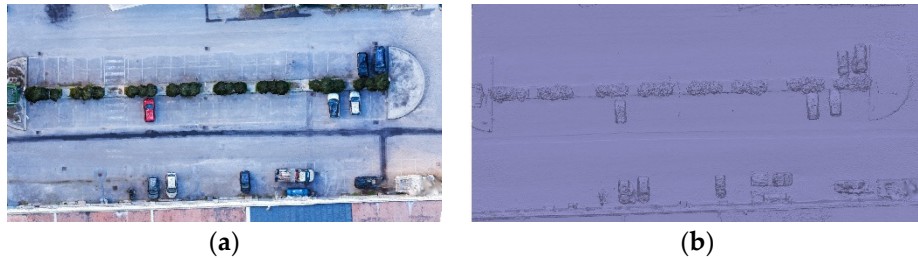

**Figure 7.** SfM reconstruction using super-resolution image dataset: (**a**) Model shaded view; (**b**) solid mesh view.

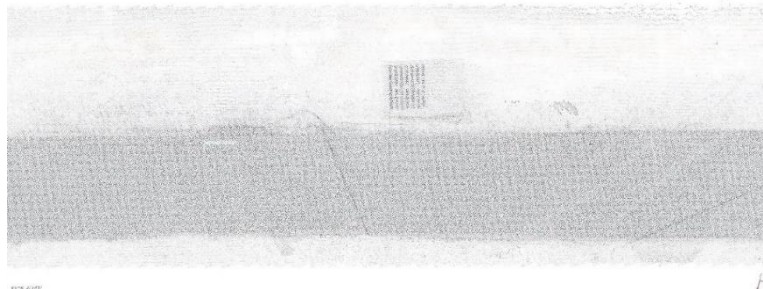

**Figure 8.** Dense cloud of low-resolution dataset.

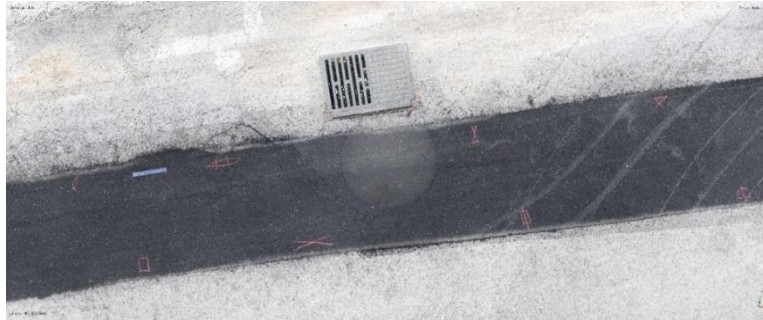

**Figure 9.** Dense cloud of super resolution dataset.

In Figures 6 and 7 the processed low and super resolution datasets are shown respectively.

In Figures 8 and 9 the results of the processing on the dense clouds of low and super resolution are presented, respectively. As output of the 3D reconstructions processed with Metashape software, the reports of acquisitions imported and processed with the SfM methodology and the calibration coefficients within the correlation matrix are shown for the Low-Resolution dataset (Tables 3 and 4) and the Super-resolution dataset (Tables 5 and 6) respectively.

**Table 3.** Summary of 3D model reconstruction from LR images.

| Number of Images: | 20 | Camera Stations: | 20 |
|---|---|---|---|
| Flying altitude | 49.7 m | Tie points: | 11.985 |
| Ground resolution: | 1.24 cm/pix | Projections: | 36.698 |
| Coverage area: | $5.01 \times 10^3$ m$^2$ | Reprojection error: | 0.823 pix |

**Table 4.** Calibration coefficients and correlation matrix related to the LR 3D model.

| | Value | Error | F | $C_x$ | $C_y$ | $K_1$ | $K_2$ | $K_3$ | $P_1$ | $P_2$ |
|---|---|---|---|---|---|---|---|---|---|---|
| F | 4295.9 | 2.1 | 1.00 | −0.32 | −1.00 | −0.18 | 0.22 | −0.28 | −0.09 | 0.06 |
| $C_x$ | 53.9991 | 0.24 | | 1.00 | 0.33 | −0.04 | 0.00 | 0.04 | 0.94 | −0.11 |
| $C_y$ | 65.7928 | 3.1 | | | 1.00 | 0.15 | −0.19 | 0.25 | 0.10 | −0.07 |
| $K_1$ | −0.015095 | 0.00022 | | | | 1.00 | −0.97 | 0.91 | −0.05 | −0.01 |
| $K_2$ | 0.030799 | 0.00075 | | | | | 1.00 | −0.98 | 0.04 | 0.05 |
| $K_3$ | −0.031978 | 0.00083 | | | | | | 1.00 | −0.01 | −0.05 |
| $P_1$ | 0.003128 | $1.9 \times 10^{-5}$ | | | | | | | 1.00 | −0.03 |
| $P_2$ | −0.00052 | $1.3 \times 10^{-5}$ | | | | | | | | 1.00 |

**Table 5.** Summary of 3D model reconstruction from SR images.

| Number of Images: | 20 | Camera Stations: | 20 |
|---|---|---|---|
| Flying altitude | 51.6 m | Tie points: | 19.855 |
| Ground resolution: | 7.11 mm/pix | Projections: | 38.519 |
| Coverage area: | $4.77 \times 10^3$ m$^2$ | Reprojection error: | 1.05 pix |

**Table 6.** Calibration coefficients and correlation matrix related to the SR 3D model.

| | Value | Error | F | $C_x$ | $C_y$ | $K_1$ | $K_2$ | $K_3$ | $P_1$ | $P_2$ |
|---|---|---|---|---|---|---|---|---|---|---|
| F | 6950.26 | 2.6 | 1.00 | −0.29 | −1.00 | −0.18 | 0.21 | −0.26 | −0.04 | 0.07 |
| $C_x$ | 85.7596 | 0.26 | | 1.00 | 0.30 | −0.07 | 0.04 | 0.00 | 0.94 | −0.13 |
| $C_y$ | 97.1246 | 3.8 | | | 1.00 | 0.14 | −0.18 | 0.24 | 0.05 | −0.07 |
| $K_1$ | −0.0151 | 0.00015 | | | | 1.00 | −0.96 | 0.91 | −0.09 | −0.03 |
| $K_2$ | 0.029503 | 0.00054 | | | | | 1.00 | −0.98 | 0.08 | 0.04 |
| $K_3$ | −0.02972 | 0.0006 | | | | | | 1.00 | −0.05 | −0.04 |
| $P_1$ | 0.003039 | $1.3 \times 10^{-5}$ | | | | | | | 1.00 | −0.04 |
| $P_2$ | −0.00049 | $9.3 \times 10^{-6}$ | | | | | | | | 1.00 |

In Figures 10–12 the results of Low-resolution and Super-resolution datasets are shown. In particular the image residuals and the camera location are represented for both dataset; in Table 7 the error estimation concerning the camera location are given.

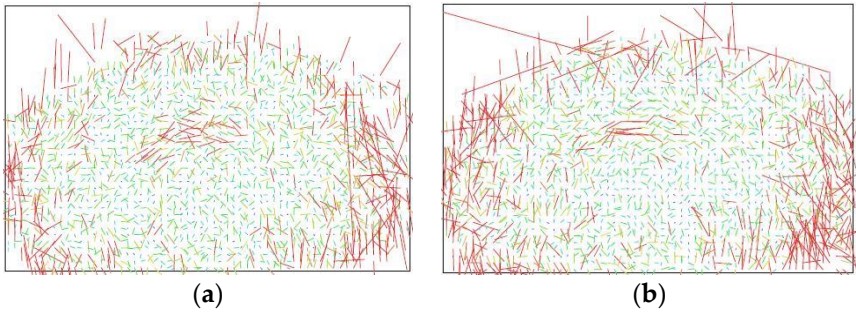

(**a**)　　　　　　　　　　　　　　(**b**)

**Figure 10.** Image residuals: (**a**) Low-resolution dataset (resolution 5568 × 3648; focal length 10.26 mm; pixel Size 2.38 × 2.38 μm); (**b**) super-resolution dataset (resolution 9000 × 5897; focal length 10.26 mm; pixel Size 1.47 × 1.47 μm).

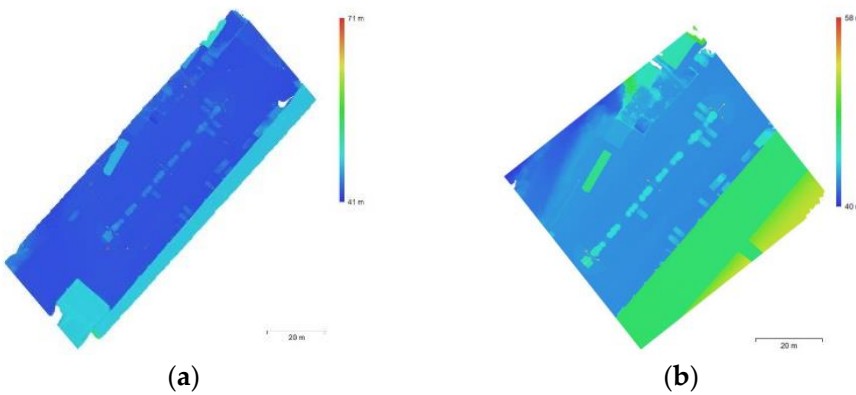

(**a**)                (**b**)

**Figure 11.** Reconstructed digital elevation model: (**a**) Low-resolution dataset (resolution 2.47 cm/pix.; point density 0.164 points/cm$^2$); (**b**) super-resolution dataset (resolution 1.42 cm/pix.; point density 0.194 points/cm$^2$).

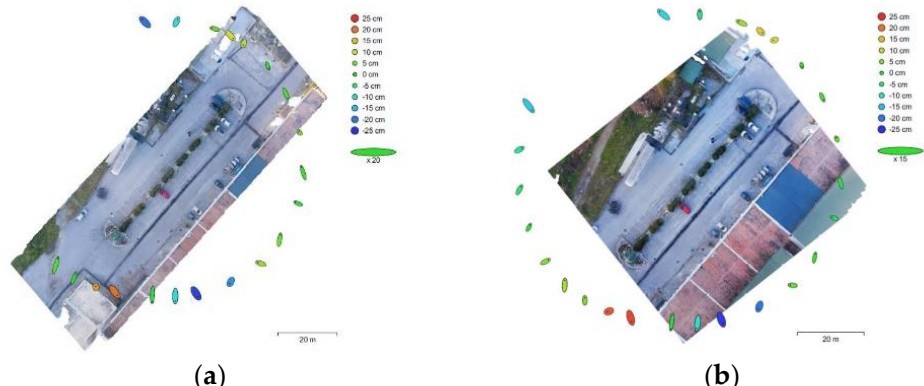

(**a**)                (**b**)

**Figure 12.** Camera locations and error estimation: (**a**) LR; (**b**) SR.

**Table 7.** Average camera location error for both SR and LR 3D models.

| X Error (cm) | Y Error (cm) | Z Error (cm) | XY Error (cm) | Tot. Error (cm) |
|---|---|---|---|---|
| 5.135658 [1] | 8.424 [1] | 7.5754 [1] | 9.8569 [1] | 12.0556 [1] |
| 1.22132 [2] | 3.8425 [2] | 2.3766 [2] | 4.7335 [2] | 5.6146 [2] |

[1] Errors from low-resolution images 3D model. [2] Errors from super-resolution images 3D model.

## 5. Results and Discussion

The optimization of the data set allowed us to carry out a very important output that can open new frontiers in road monitoring field.

The alignments of the low and super resolution cloud with the ground truth one, show how the metric accuracy increases as the RMSE value decreases, that is, it increases with the processing of the data set with the super resolution images

It is important to underscore that without any visual adjustment it would be impossible to get the alignment between the low resolution cloud with the ground truth one due the difficulty in recognizing the homologous markers [74]. Figures 13 and 14, show the merged dense clouds of the low and super resolution dataset of the road pavement surface with the ground truth one.

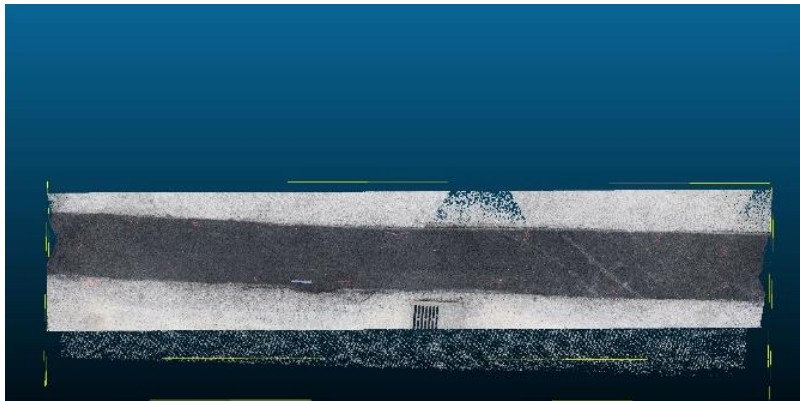

**Figure 13.** Merge cloud of low-image data set dense cloud and ground truth dense cloud.

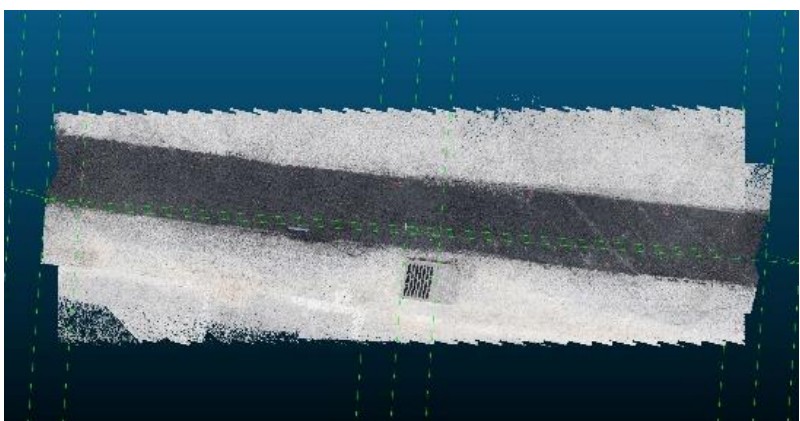

**Figure 14.** Merge cloud of super resolution-image data set dense cloud and ground truth dense cloud.

The reading of the values of the mean square deviation of the comparisons between the two clouds, Low and Super clouds respectively, shows that the value in the case of the dataset originating from the flight, the mean square deviation exceeds the value of 10 cm and, along the entire surface of the comparison the values drop slightly (red area in the diagram) (Figure 15).

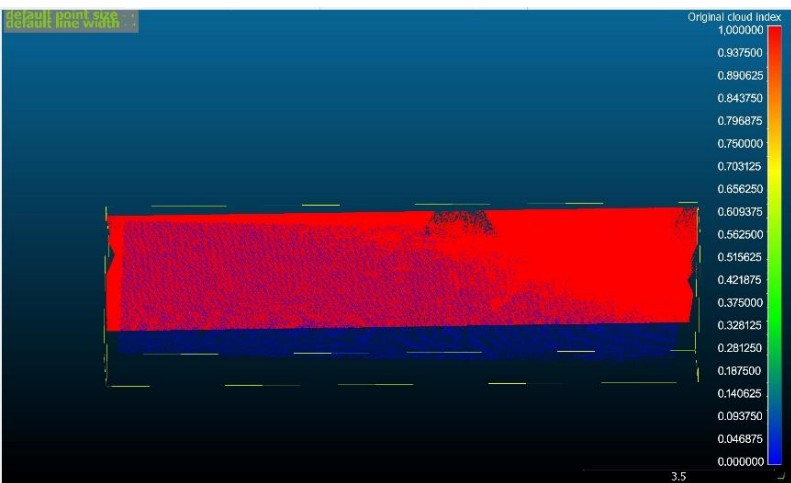

**Figure 15.** RMSE histogram in cm for Low dense cloud to ground truth cloud.

The merge clouds referring to the different comparison underline that the merge dense cloud obtained from the super-resolution is closer the truth one, more than the Lower-resolution merge dense cloud. In fact, the result of the comparison with the Super

resolution data set shows a very different result from the previous one (blue area in the diagram) (Figure 16). The range of the value of the RMS goes from 0 to 1.5 cm.

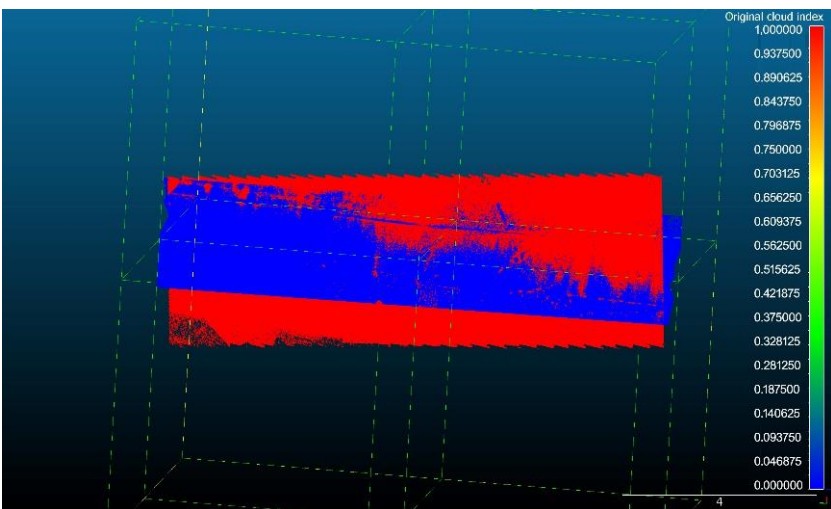

**Figure 16.** RMSE histogram in cm for super resolution dense cloud to ground truth cloud.

## 6. Conclusions

This study was conducted, principally, to show the applicability of the SRa to images acquired by a drone 30 m above the pavement. This height was chosen as it is likely to be sufficient to avoid most physical obstacles in real world applications. Naturally, images taken from this height are more susceptible to noise and other interference and, resultantly, the fine detail which is required to conduct an accurate analysis is often lost or obscured. The SRa is designed to digitally enhance the quality of low-resolution images and the results presented demonstrated this algorithm's applicability in increasing the accuracy of the resultant 3D model. This was demonstrated through a dramatic reduction in the RMSE from in excess of 10 cm to between 0 and 1.5 cm.

The results achieved demonstrated that, in case of low-resolution images, is opportune to increase the quality of the source image using the super-resolution image software to achieve at a better-quality model. In this investigation the performance of the Super-resolution image was evaluated, not so much for the recognition of the images, as for the construction of the 3D model from images. The proposed methodology envisaged two main phases: the first one to process the low-resolution images in Super-resolution and the second one to process the data set to 3D reconstruction. Finally, the dense clouds have been compared to verify the quality of the 3D model information. In other words, the comparison was conduct from the dense cloud generated from the data set processed from low to super resolution. The values of RMS achieved, demonstrate that there is an RMS $\geq$ 10 cm for the comparison of the pavement distress low dense cloud and a $0 \leq RMS \geq 1.5$ cm for the super resolution dense cloud. In other words, in case of low-resolution data set, the super-resolution image processing, improve the quality of the 3D model (Figure 17).

The metric accuracy changes its value according to the matrix transformation of the data set images. In the path of the super resolution algorithm the matrix transformation achieves different value of accuracy. Figure 17 shows as the geometric accuracy values of super resolution data set is always better than the original one.

In future studies, the authors intend to more deeply investigate the effects of the SRa on low-resolution images to create accurate 3D models from which distresses in road pavements can be easily and accurately detected. This study serves as a proof of concept tying in techniques from multiple scientific disciplines, the results presented are promising and therefore ripe for further exploration.

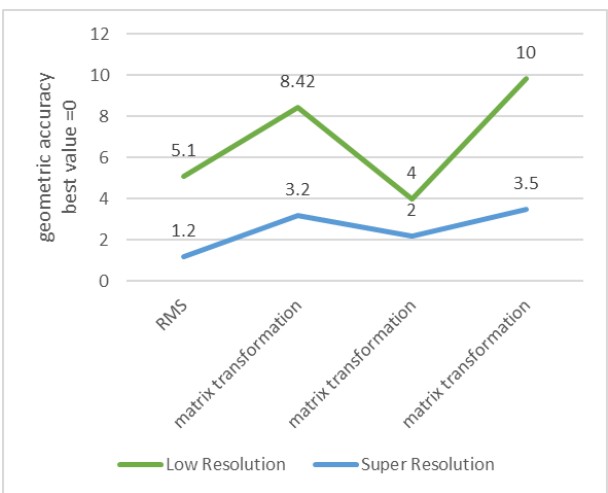

**Figure 17.** RMS [m] and matrix transformation for both low and super resolution images.

**Author Contributions:** Conceptualization, L.I.; data curation, L.I.; formal analysis, F.A.; funding acquisition, G.D.M.; investigation, L.I., F.A. and G.D.M.; methodology, L.I.; resources, F.A. and M.Z.U.; supervision, L.I. and G.D.M.; validation, L.I., G.D.M. and M.Z.U.; writing—original draft, L.I. and F.A.; Writing—review & editing, F.A. All authors have read and agreed to the published version of the manuscript.

**Funding:** This research has been produced with the financial assistance of the European Union under the ENI CBC Mediterranean Sea Basin Program, for Education, Research, technological development and Innovation, under the grant agreement n.28/1682.

**Institutional Review Board Statement:** Not applicable.

**Informed Consent Statement:** Not applicable.

**Data Availability Statement:** Not applicable.

**Acknowledgments:** This paper has been supported by the ENI CBC Mediterranean Sea Basin Program, for Education, Research, technological development and Innovation.

**Conflicts of Interest:** The authors declare no conflict of interest.

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
