# Peer review of "Super-Resolution Images Methodology Applied to UAV Datasets to Road Pavement Monitoring"

_drones, doi:10.3390/drones6070171_

Round 1

Reviewer 1 Report

Dear authors

Congratulations on the article, Super-Resolution images methodology applied to UAV datasets to road pavement monitoring , which is well presented, but needs revisions in some sections of the text, as follows:

Line 108 – sigma is the noise, and D what is its meaning?

Line 114 – high resolution images: Xi in place of X.

Line 119 - Equation (3) is confusing.

Line 135 – The neural network training parameters were provided, such as minimization of the loss function, learning rate, filter weights of each layer, interaction layers, but what was the neural network used?

Line 137 – Mention the algorithm used.

Line 293 – Table 5 presents mean values of errors. What is the value of the standard deviation of the mean values presented? What was the sample size? What is the statistical significance of the data presented (p-value).

Line 335 - A week point in this paper is the conclusion, it must be improved. It is suggested to expand the comments in the results.

Best regards

Author Response

Thank you for taking the time to review our article, we greatly appreciate the time taken from your busy schedule to improve the quality of our article. We strongly believe in the value of peer review and from your responses it is clear that you have read our research with care and attention. We hope that at some point in the future, we may be able to return the favour. 

In the updated manuscript you will find that your suggestions have been incorporated and the specific points mentioned have either been changed or expanded upon. Changes have also been made to the English as requested.

Warmest Regards

Reviewer 2 Report

The argument of the paper is really interesting, the research methodology is quite clear, and the achievements are excellent.

More pictures and drawings could enhance the text comprehension, especially in the methodology and investigation paragraphs.

Author Response

(The authors gave the same response as above.)

Reviewer 3 Report

The goal of this paper is to evaluate whether applying Super-resolution image techniques on aerial photographs from drones improves the quality of the reconstructed scene. The structure of the article requires heavy modification, as parts of section 3 (methodology) and 4 (investigation) are better suited in the related work section, which in turn is poor.

The authors are applying the SR methodology using commercial software. Therefore, the included novelty is limited. Regarding the case study, the authors collect two datasets from a single topographic area, making the results difficult to generalize. 

Language needs improvement to enhance the readability and the comprehension of the article.

Overall, the paper touches an important problem but the novelty introduced is rather limited. Authors are encouraged to extend the experiments with public datasets  and a more thorough exploration of the SR implementations.

Author Response

Thank you for taking the time to review our article, we greatly appreciate the time taken from your busy schedule to improve the quality of our article. We strongly believe in the value of peer review and from your responses it is clear that you have read our research with care and attention. We hope that at some point in the future, we may be able to return the favour. 

Regarding your suggestion on the limited novelty of the paper, I would like to outline one point which has now been incorporated in the conclusions of the article. 
The paper's goal was to show the applicability of a method to enhance the quality of low resolution images using the Super-Resolution algorithm. This algorithm was not developed with the express intent to improve the accuracy of pavement distress detection using an UAV. As you stated the novelty presented is, in some ways, limited, however, it is our belief that this is a novel application which brings together tools, techniques, and methodologies from multiple disciplines. In that regard, it is our belief that presenting this application and showing the increased accuracy that is facilitated through the use of this algorithm, is valid. Furthermore, drawing attention to the Super-Resolution algorithm is of demonstrable scientific benefit. It is our belief that our goal as academics should not simply be to increase the wealth of scientific knowledge but also to influence industry to adopt more effective methodologies; to this aim, it is important to clearly and concisely demonstrate concepts using case studies which are accessible. We hope that you agree with our assessment that our research has the potential to make the use of UAVs as a tool to conduct low-cost analysis of the condition of road pavement.

I would like to end, however, by agreeing that your suggestion is sound and may serve as a future study to show the applicability of our methodology on alternative datasets. 

Thank you again for your time and consideration, I hope that you agree with the argument outlined above,

Warmest Regards,

Reviewer 4 Report

The description of related works needs to be extended. The methodology section is very general. The whole algorithm of image processing should be thoroughly described. The next section includes a short description of commercial procedures used for image conversion but it is necessary to state the routines and their parameters. The investigation parts show the characteristic examples of road pavement distress but in case study section there is no evidence of such artifacts. The SfM reconstruction was described too briefly. The description of Figures 9-11 and Tables 2-5 in Chapter 4 was omitted by the authors. The numbering of some chapters and figures is repeated. Section Results and Discussion contains a non-quantitative comparison of reconstructions without details. The presented conclusions are not clearly confirmed in the conducted research. Figure 16, which could confirm the improvement in the quality of the reconstruction, is completely unclear.

Author Response

Thank you for taking the time to review our article, we greatly appreciate the time taken from your busy schedule to improve the quality of our article. We strongly believe in the value of peer review and from your responses it is clear that you have read our research with care and attention. We hope that at some point in the future, we may be able to return the favour. 

In the updated manuscript you will find that your suggestions have been incorporated and the specific points mentioned have either been changed or expanded upon. Changes have also been made to the English as requested.

Kind Regards

Round 2

Reviewer 3 Report

Dear authors. I appreciate the effort that has been paid to improve the structure and the quality of the document. However, I still believe that more novelty is required for a journal publication even if it is application-related. Therefore, to my regret, I will stay with my previous decision.

Reviewer 4 Report

I have no further remarks.